# MultipleTesting.com: A tool for life science researchers for multiple hypothesis testing correction

Otília Menyhart[1,2], Boglárka Weltz[2,3], Balázs Győrffy[1,2,4]*

1 Department of Bioinformatics, Semmelweis University, Budapest, Hungary, 2 Research Centre for Natural Sciences, Cancer Biomarker Research Group, Institute of Enzymology, Budapest, Hungary, 3 A5 Genetics Ltd, Und, Hungary, 4 2nd Department of Pediatrics, Semmelweis University, Budapest, Hungary

* gyorffy.balazs@med.semmelweis-univ.hu

**Data Availability Statement:** All relevant data are within the paper and S1–S3 Scripts.

**Funding:** The research was financed by the 2018-2.1.17-TET-KR-00001, 2020-1.1.6-JÖVŐ-2021-00013 and 2018-1.3.1-VKE-2018-00032 grants

## Abstract

Scientists from nearly all disciplines face the problem of simultaneously evaluating many hypotheses. Conducting multiple comparisons increases the likelihood that a non-negligible proportion of associations will be false positives, clouding real discoveries. Drawing valid conclusions require taking into account the number of performed statistical tests and adjusting the statistical confidence measures. Several strategies exist to overcome the problem of multiple hypothesis testing. We aim to summarize critical statistical concepts and widely used correction approaches while also draw attention to frequently misinterpreted notions of statistical inference. We provide a step-by-step description of each multiple-testing correction method with clear examples and present an easy-to-follow guide for selecting the most suitable correction technique. To facilitate multiple-testing corrections, we developed a fully automated solution not requiring programming skills or the use of a command line. Our registration free online tool is available at www.multipletesting.com and compiles the five most frequently used adjustment tools, including the Bonferroni, the Holm (step-down), the Hochberg (step-up) corrections, allows to calculate False Discovery Rates (FDR) and q-values. The current summary provides a much needed practical synthesis of basic statistical concepts regarding multiple hypothesis testing in a comprehensible language with well-illustrated examples. The web tool will fill the gap for life science researchers by providing a user-friendly substitute for command-line alternatives.

## Introduction

Technological innovations of the past decades enabled the concurrent investigation of complex issues in biomedical sciences and increased their reliance on mathematics. For example, high-throughput omics-based technologies (e.g., genomics, transcriptomics, proteomics, metabolomics) involving hundreds or thousands of markers offer tremendous opportunities to find associations with the phenotype. Analyzing a massive amount of data by simultaneous statistical tests, as in genomic studies, is a double-edged sword. Conducting multiple comparisons

and by the Higher Education Institutional Excellence Programme (2020-4.1.1.-TKP2020) awarded to B. Gy, of the Ministry for Innovation and Technology in Hungary, within the framework of the Bionic thematic programme of the Semmelweis University. The funders had no role in study design, data collection and analysis, decision to publish, or preparation of the manuscript. The authors wish to acknowledge the support of ELIXIR Hungary (www.elixir-hungary.org). The commercial company A5 Genetics Ltd, Hungary provided support in the form of salaries for one author [B. W.], but did not have any additional role in the study design, data collection and analysis, decision to publish, or preparation of the manuscript. The specific role of this author is articulated in the 'author contributions' section.

**Competing interests:** The authors B. Gy. and O. M declare no potential conflicts of interest. The author B. W. received salary from the commercial company A5 Genetics Ltd, Hungary. This does not alter our adherence to PLOS ONE policies on sharing data and materials.

increases the likelihood that a non-negligible proportion of associations will be false positives while also increases the number of missed associations (false negatives) [1]. The problem is not exclusive to biomedical sciences; researchers from nearly all disciplines face the problem of simultaneous evaluation of many hypotheses, where the chance of incorrectly concluding at least one significant effect increases with each additional test. Drawing valid conclusions require taking into account the number of performed statistical tests and adjusting the statistical confidence measures.

Several strategies exist to overcome difficulties when evaluating multiple hypotheses. Here we review the most frequently used correction approaches, illustrate the selected methods by examples, and provide an easy-to-follow guide to facilitate the selection of proper strategies. We also summarize the basic concepts of statistical tests, clarify conceptual differences between exploratory and confirmatory analyses, and discuss problems associated with the categorical interpretation of p-values.

To facilitate the interpretation of multiple hypothesis tests, we established a quick and user-friendly solution for automated multiple testing correction that does not require programming skills or the use of a command line. Our tool available at www.multipletesting.com allows choosing from the most frequently used multiple-testing correction methods, including the Bonferroni, the Holm (step-down), the Hochberg (step-up) adjustments, calculation of False Discovery Rates (FDR), and q-values.

## Basic concepts of statistical inference

In a formal scientific method, the null hypothesis (H0) is the one we are seeking to disprove, representing no differences in measurements between two groups (e.g., there is no effect of a given gene on a trait of interest). The null hypothesis is compared with a statistical test to the alternative hypothesis (H1), the antithesis of the null, assuming differences between groups (e.g., there is an association between a gene and a phenotypic trait). The procedure results in a statistical confidence measure, called a p-value, compared to the level of significance, α. Thus the p-value represents how extreme the data are while the α determines how extreme the data must be before the null hypothesis is rejected. When p is smaller than the confidence threshold α, the null hypothesis is rejected with a certain confidence, but the rejection does not "prove" the alternative hypothesis. If the p-value is higher than α, the null hypothesis is not rejected, although it does not mean the null is "true", only there is not enough evidence against it.

Four outcomes are possible of a statistical test: the test rejects a false null hypothesis (true positives), the test rejects a true null (type I error or false positives), the test does not reject a true null (true negatives), or the test does not reject a false null (type II error or false negatives). The level of significance, α, controls the level of false positives. Historically, α values have been set at 0.05 [2]; this is the per comparison error rate. One can set α values at a more conservative level to further decrease the type I error, such as at 0.01. However, there will be a corresponding increase in a type II error, the failure to detect a real effect, therefore it is advisable to strike a balance between the two types of errors.

Instead of representing the metric of "truth" or "significance", in reality, a p-value of 0.05 means that there is 5% chance to get the observed results when the null hypothesis is true, when statistical results may not be translated into biologically relevant conclusions. For example, if we measure 20 different health parameters at p = 0.05 in a patient where all the nulls are true, one out of 20 will statistically deviate from the normal range, but without biological relevance (false positive). Following the same logic, when 20,000 genes are analyzed between two samples, the expected number of false positives increases to a substantial 1000. The elevated number of simultaneous statistical tests increases the danger that the number of irrelevant

false positives exceeds the number of true discoveries; therefore, multiple-testing correction methods are required.

## Misinterpretation of statistical significance

A great concern is that the p-value is frequently treated as a categorical statistical measure, also being reflected in reporting of the data: instead of being disclosed with precision (e.g. p = 0.081 or 0.8), p-values are described as categorical inequalities around an arbitrary cut-off (p > 0.05 or p < 0.05). The greatest concern is that results below the statistical threshold are frequently portrayed as "real" effects, while statistically non-significant estimates are treated as evidence for the absence of effects [3]. Such dichotomization represents a severe problem in the scientific literature: findings interpreted as "real" are frequently inflated in reports, privileged in publications, and even may bias the initial choice of data and methods to reach the desired effects [3]. Nevertheless, p-values are not to be banned because unfounded claims could promote bias, but more stringent thresholds are needed, and clear rules should be set before data collection and analysis [4].

## Conceptual differences between confirmatory and exploratory hypothesis testing

One always must consider the test statistics when interpreting p-values. If the sample size is too large, small and irrelevant effects might produce statistically significant results. Small sample size or large variance may, on the contrary, can render a remarkable effect to be insignificant.

Another principle is to differentiate between an exploratory and confirmatory hypothesis test and the resulting p-values. As their name suggests, exploratory analyses explore novel information within a data set to establish new hypotheses and novel research directions. To fulfill this purpose, all comparisons should be tested, followed by an appropriate adjustment of p-values. Consequently, exploratory analyses are suitable to generate hypotheses but do not "prove" them.

Confirmatory analyses, on the contrary, are testing "a priori" identified, specific hypotheses, intending to confirm or reject a limited number of clearly articulated assumptions, where significance levels are also established beforehand [5]. For example, to promote best practices in clinical sciences, protocols and hypotheses of clinical studies are required to be registered ahead, e.g., at ClinicalTrials.gov. When testing multiple hypotheses, the p-values should be corrected in confirmatory analyses as well.

To choose a suitable correction method, one must consider the exploratory or confirmatory nature of the conducted statistical tests. A decision tree depicted in **Fig 1** helps to guide and refine the selection of appropriate correction tools for exploratory vs. confirmatory types of statistical analyses. However, one must also consider a wide range of studies represented by a mixture of exploratory or confirmatory analyses.

## Methods of multiple-testing corrections

The topic is far-reaching and rapidly proliferating to be covered in its entirety, and comparing technical parts of actual statistical methods is beyond our intended scope. Here we aim to introduce the most extensively utilized methods.

The common denominator across the presented methods dealing with multiplicity is that all of them reject the null hypothesis at the smallest p-values; still, there is a difference in the number of rejected hypotheses.

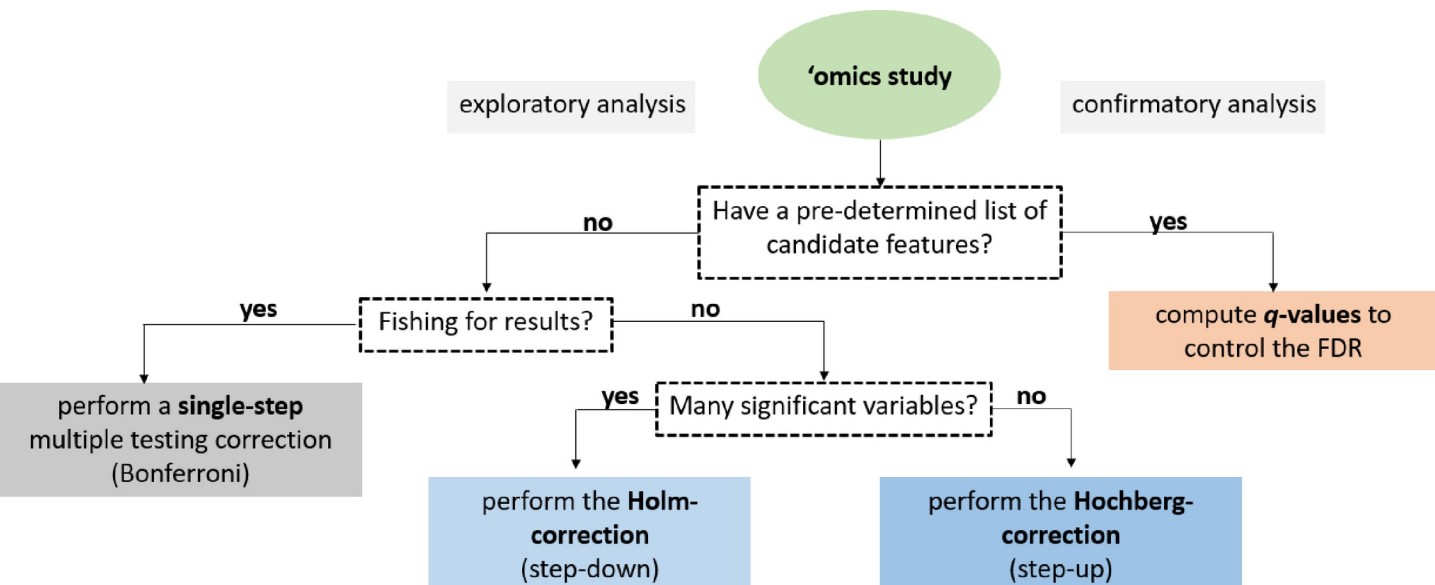

**Fig 1. A decision tree to facilitate the selection of suitable methods for multiple testing correction.** The initial decision relies upon the statistical analysis's exploratory or confirmatory nature, while subsequent steps narrow down the list of appropriate methods. In some cases, though, studies are based on a mixture of exploratory or confirmatory analyses.

## Single-step methods for multiple-testing correction

In single-step corrections, equivalent adjustments are made to each p-value. The simplest and most widely used correction method is the Bonferroni-procedure, which accounts for the number of statistical tests and does not make assumptions about relations between the tests [6–8]. The Bonferroni-procedure aims to control the family-wise error rate (FWER), referring to the probability of committing at least one type I error amongst multiple statistical analyses. Family-wise error rates thus allow very few occurrences of false positives (**Table 1**).

There are two approaches to calculate the adjusted p-values with the Bonferroni-procedure. According to the first method, one may divide the per analysis error rate by the number of comparisons ($\alpha/n$). Only p-values smaller than the adjusted p-value would be declared statistically significant. For example, if we have five measurements and $\alpha = 0.05$, only p-values $< 0.01$ (five divided by 0.05) would be reported significant.

According to the second method, the p-value of each test ($p_i$) is multiplied by the number of performed statistical tests (n): $np_i$. If the adjusted p-value is lower than the significance level, $\alpha$ (usually 0.05), the null hypothesis will be rejected, and the result will be significant (**Table 1**). For example, if the observed p-value is 0.016 and there are 3 measurements with $\alpha = 0.05$, the adjusted p-value would be $0.016 * 3 = 0.048$, which is less than 0.05, and the null hypothesis would be rejected.

Another well-known one-step correction method is the Sidak-correction [9, 10] (for the formula and explanation, see **Table 1**). The method assumes that the performed tests are independent of each other, thus may not be appropriate for every situation. Moreover, when facing a large number of tests, the results of the Sidak-correction can be reasonably approximated with the Bonferroni-adjustment [11].

The Bonferroni-adjustment works well in settings where the number of statistical tests does not exceed a couple of dozens to a couple of hundreds, as in candidate gene studies or genome-wide microsatellite scans, respectively. Nevertheless, the Bonferroni-correction is the most

Table 1. Formulas and explanations of statistical concepts.

| Statistical concept | Formula | Explanation |
|---|---|---|
| Family-wise error rate | $\alpha FW = 1- (1- \alpha PC)C$ | where C refers to the number of comparisons performed, and $\alpha PC$ refers to the per contrast error rate, usually 0.05 |
| Bonferroni correction | $p'_i = np_i \leq \alpha$ | the p-value of each test ($p_i$) is multiplied with the number of performed statistical tests (n). If the corrected p-value ($p'_i$) is lower than the significance level, $\alpha$ (usually 0.05), the null hypothesis will be rejected and the result will be significant |
| Sidak-correction | $p'_i = 1- (1—p_i)n \leq \alpha.$ | where $p_i$ refers to the p-value of each test, and n refers to the number of performed statistical tests (n) |
| False discovery rate | $FDR = E [V / R]$ | where E refers to the expected proportion of null hypotheses that are falsely rejected (V), among all tests rejected (R), thus it calculates the probability of an incorrect discovery. |
| Storey's positive FDR | $FDR (t) = (\pi 0 \times m \times t) / S(t)$ | where t represents a treshold between 0 and 1 under which p-values are considered significant, m is the number of p-values above the treshold ($p_1, p_2, p_m$), $\pi 0$ is the estimated proportion of true nulls ($\pi 0 = m0 / m$) and S(t) is the number of all rejected hypotheses at t |
| Benjamini and Yekutieli | $FDR_i \leq (n \times p_i)/ (nR_i \times c(n))$' | where c(n) is a function of the number of tests depending on the correlation between the tests. If the tests are positively correlated, c(n) = 1 |
| Proportion of false positives (PFP) | $PFP = E (V) / E (R)$ | where E refers to the expected proportion of null hypotheses that are falsely rejected (V), among all tests rejected (R). V and R are both individually estimated |
| q-value | $q(p_i) = min FDR (t)$ | the q-value is defined as the minimum FDR that can be achieved when calling that "feature" significant |

stringent method with the major disadvantage of over-adjusting p-values, erroneously increasing the probability of false negatives, and overlooking positive signals when evaluating a large number of tests. For example, in genomic studies testing for 40,000 genes, the adjusted p-value would decrease from p = 0.05 to the impossibly low p = 0.00000125. Novel statistical approaches are available to avoid over-adjustment.

## Sequential methods for multiple-testing correction

The **Holm-correction**, also called the step-down method, is very similar to the Bonferroni-adjustment with a similar family-wise error rate, but is less conservative and applied in stages [12]. The p-values are ranked from the smallest to largest; then, the smallest p-value is multiplied by the number of all statistical tests. If the adjusted p-value is less than $\alpha$ (usually 0.05), the given test would be declared statistically significant and removed from the pool of investigated p-values. The sequence will be continued in this fashion (with the adjustment of n-1 corresponding to the rank) until no gene is significant.

For example if we conducted n = 500 statistical tests with the three smallest p-values being 0.00001, 0.00008, 0.00012, and $\alpha$ = 0.05, the following adjustments are concluded:

Rank#1: 0.00001 * 500 = 0.005, 0.005 < 0.05, the test is significant, reject the hull hypothesis

Rank#2: 0.00008 * 499 = 0.0398, 0.0398 < 0.05, the test is significant, reject the hull hypothesis

Rank#3: 0.00012 * 498 = 0.0596, 0.0596 > 0.05, the test is <u>not significant</u>, and none of the remaining p-values will be significant after correction.

The **Hochberg-correction**, also called the step-up method, is based on a reverse scenario when the largest p-value is examined first. Once a significant p-value is identified, all the remaining smaller p-values would be declared significant [13]. For example, if n = 500, the

largest p-values are 0.0015, 0.00013, 0.00001, and α = 0.05, the following adjustments are concluded:

Rank#1: 0.0015 * 500 = 0.75, 0.75 > 0.05, the test is not significant

Rank#2: 0.00013 * 499 = 0.0649, 0.0649 > 0.05, the test is not significant

Rank#3: 0.00001 * 498 = 0.0498, 0.0498 < 0.05, the test is significant, reject the hull hypothesis, and all of the remaining p-values will be significant after correction.

Generally, the Hochberg-correction retains a larger number of significant results compared to the one-step and Holm-corrections. The Holm- and Hochberg-corrections are beneficial when the number of comparisons is relatively low while the effect rate is high, but are not appropriate for the correction of thousands of comparisons.

## Controlling the False-Discovery Rate (FDR)

The widespread application of RNA-seq and microarray-based gene expression studies have greatly stimulated research on the problem of massive hypothesis testing. Controlling the **False-Discovery Rate** described by Benjamini-Hochberg (1995) provides a right balance between discovering statistically significant effects and limitations by false positives and is the least stringent of all the included methods [14]. The FDR is particularly appropriate for exploratory statistical analyses testing multiple hypotheses simultaneously, used extensively in fields like genetics where some true effects are expected to be seen among the vast number of zeros, such as when assessing treatment effects on differential gene expression.

The FDR is calculated as the expected proportion of null hypotheses falsely rejected among all tests rejected, thus calculating the probability of an incorrect discovery. To clarify the distinction between the error rate and FDR, the error rate of 0.05 means that 5% of truly null hypotheses will be called significant on average. In contrast, FDR controlled at 5% means that out of 100 genes considered statistically significant, five genes will be truly null on average.

In practice, the procedure is based upon the ranking of p-values in ascending order after which each individual p-value's Benjamini-Hochberg critical value is calculated by dividing the p-value's individual rank with the number of tests, multiplied by the False Discovery Rate (a percentage chosen by the researcher) (**Table 1**). For example, in case of n = 100 statistical tests, for the smallest p-value 0.0001, the critical value at FDR = 5% is calculated as 1/100 * 0.05 = 0.0005. Since the p-value 0.0001 < 0.0005, the test is significant at 5% FDR.

For the second smallest p-value, the critical value would be calculated as 2/100 * 0.05 = 0.01. With the Benjamini-Hochberg procedure, we are searching for the highest p-value that is smaller than the critical value. All the p-values lower than the identified p-value would be considered significant.

Various alternative methods have been developed to provide a more precise estimation of the FDR [15]. A modification has been proposed by Storey, called the positive FDR (pFDR), assuming that at least one positive finding has occurred [16]. Storey's method adopts a rank scheme similar to the Benjamini-Hochberg procedure, except it introduces the estimated proportion of true nulls ($\pi0$) (for the formula, see **Table 1**). $\pi0$ can be estimated as 2/N times the number of p-values greater than 0.5 [17] or twice the average of all p-values [18, 19]. The pFDR controls the expected proportion of false positives in each experiment, and since the probability of R > 0 is ~ 1, in most genomics experiments, pFDR and FDR are very similar.

The Benjamini-Hochberg method is sufficient for most cases, especially when tests are independent and p-values are uniformly distributed. Simulations suggest that multiple testing correction methods perform reasonably well even in the presence of weak positive

correlations, which is common in genetic studies [20, 21]. Positive correlation among tests is also a frequent condition in ecological studies. Benjamini and Yekutieli developed a refinement that controls the FDR under arbitrary dependence assumptions by introducing a function of the number of tests depending on the correlation between tests [21] (**Table 1**).

When the assumption of independence among p-values is not fulfilled, another method is available to control the proportion of false positives (PFP) among all positive test results [22]. To estimate PFP, it is necessary to calculate the proportion of true nulls. The method is similar to FDR and pFDR calculations, but PFP does not depend on the correlation among tests or the number of tests (**Table 1**) [22].

## The local False Discovery Rate

The FDR introduced by Benjamini and Hochberg (1995) is a global measure and can not assess the reliability of a specific genetic marker. Contrarily, the local FDR can quantify the probability of a given null hypothesis to be true by taking into account the p-value of individual genetic markers, thus assessing each marker's significance. The method is particularly suitable if the intentions are to follow up on a single gene. However, the method requires an estimation of true nulls ($\pi 0$) and the distribution under the alternative hypothesis [23], and in general, it is quite difficult to estimate precisely. Additional developments enhancing the understanding and interpretability of the original FDR control are available in [24].

## The q-value

The q-value is the FDR analog of the p-value in multiple hypothesis testing; adjusted p-values after FDR corrections are actually q-values. The q-value offers a measure of the strength of the observed statistic concerning the FDR; it is defined as the minimum FDR at which the test would be declared significant; in other words, it provides the proportion of significant features that turn out to be false leads (**Table 1**) [16, 17]. The estimated q-value is a function of the p-value for that test and the distribution of the entire set of p-values from the entire family of simultaneous tests [16]; thus, q-values are increasing according to p-values. The q-value has been extensively used in the analysis of microarray and sequencing data.

Calculating false positives according to p-values considers all statistical tests, while q-values take into account only tests with q-values less than the chosen threshold. The concept is illustrated with the following scenario: in a genomic study with 5000 statistical tests, geneX has a p-value of 0.015 and a q-value of 0.017. In the dataset, there are 500 genes with p-values of 0.015 or less. According to the 1.5% false-positive rate, 0.015 * 5000 = 75 genes would be expected to be false positives. At q = 0.017, 1.7% of genes with p-values as small or smaller as geneX's will be categorized as false positives; thus, the expected number of false positives is 0.017 * 500 = 8.5, which is much lower than the predicted 75.

## Modern FDR-controlling methods

Methods involving the Benjamini-Hochberg FDR control and Storey's *q* work solely off the *p* values and assume that all statistical tests have the same power to detect discoveries. However, individual tests may differ in their statistical properties due to differences in sample size, signal-to-noise ratio, or underlying biology, and some tests may have greater power for detection. Since the seminal paper by Benjamini and Hochberg (1995) [14], the development of multiple hypothesis testing was mainly motivated by the increasing availability of high-throughput technologies in many areas of science. When the number of statistical tests is in the thousands, methods assigning weight to individual *p*-values improve the power [25]. Nevertheless, in the era of modern omics—genomics, epigenomics, proteomics, metabolomics, etc.- the problem

of multiple hypothesis testing is expanding when weights need to be allocated in a data-driven manner.

Modern FDR-controlling methods include an informative covariate, encoding contextual information about the hypotheses to weigh and prioritize statistical tests, with an ultimate goal to increase the overall power of the experiment. Various covariate-aware FDR methods have been developed to provide a general solution for a wide range of statistical problems. A recent systematic evaluation summarized the merits of two classical and six modern FDR-controlling methods on real case studies using publicly available datasets [26]. The novel methods included the conditional local FDR [27], the FDR regression [28], independent hypothesis weighting [29], adaptive shrinkage [30], the method based on Boca and Leek's FDR regression [31], and the adaptive p-value thresholding procedure (AdaPT), the latter marrying machine learning methods with multiple testing solutions [32, 33]. The novel methods were modestly more powerful than the Benjamini-Hochberg FDR and Storey's *q*, and their performance depended on the particular settings [26]. Another novel covariate-aware FDR method (AdaFDR) was able to adaptively learn the optimal p-value from covariates, resulting in more associations than the Benjamini-Hochberg procedure at the same FDR [34].

### Additional methods for multiple hypothesis testing

Besides the introduced tools, additional complex strategies are available that require extensive skills in both statistics and programming. Excellent comprehensive reviews describe additional concepts, such as guidelines for large-scale genomic studies [35] or proteomics and metabolomics applications [36].

While most research has been dedicated to continuous data, high dimensional count and binary data are also common in genomics, machine learning, and imaging, such as medical scans or satellite images. An excellent summary discusses the extension of concepts in multiple testing for high dimensional discrete data, such as false discovery rate estimation to the discrete setting [37]. In discrete test statistics, methods for estimating the proportion of true null hypotheses are particularly useful for high-throughput biological RNA-seq and single-nucleotide polymorphism data [38].

Modern biopharmaceutical applications also utilize multiplicity control, such as clinical trials involving interim and subgroup analyses with multiple treatment arms and primary and secondary endpoints. For example, when aiming to find dose effects, hypotheses are *a priori* ordered based on their clinical importance. Clinical trials may be hierarchically ordered with logical relations: the secondary hypothesis can only be tested if the primary hypothesis is significant; thus, the latter acts as a gatekeeper for the overall family of hypotheses. Various gatekeeping procedures exist to handle multiple hierarchical families of hypotheses [39]. Another alternative for performing Bonferroni-type sequential testing is the graphical approach allowing the visualization of multiple hypotheses via weighted graphs [40].

For multiple testing procedures, R packages are available via CRAN (36) or Bioconductor [41], particularly the packages multtest [42], and qvalue [43] (Bioconductor), fdrtool [44], and mutoss [45] (CRAN). Moreover, all listed novel FDR-controlling methods, but one (local FDR), are available as R packages [26]. As their implementation requires programming skills and a command line, there remains a demand for user-friendly automated tools.

### To choose multiple testing correction methods

Multiple hypothesis testing corrections help to avoid unjustified "significant" discoveries in many fields of life sciences. The question remains: which method should be used for a particular analysis? It depends on the trade-off between our tolerance for false positives and the

benefit of discovery. The exploratory and confirmatory nature of the planned research is also indicative. Asking the right questions before the analysis will narrow down the number of possibilities, as illustrated with a decision tree in **Fig 1**.

## Conclusions

Controlling for false positives is particularly relevant in biomedical research. Applications include the selection of differentially expressed genes in RNA-seq or microarray experiments, where expression measures are associated with particular covariates or treatment responses; genetic mapping of complex traits based on single nucleotide polymorphisms when evaluating the results of genome-wide association studies (GWAS); scanning the genome for the identification of transcription factor binding sites or searching a protein database for homologs, etc. [35, 46]. Adjustment for multiple testing is also standard procedure in economics [47], psychology [48], and neuroimaging [49], among others.

Choosing and successfully conducting appropriate multiple testing corrections may require extensive literature and background investigations with a steep learning curve. The use of adjustment methods improves the soundness of conclusions, although they are still underutilized by many life science researchers. We hope the current summary provides a much-needed practical synthesis of basic statistical concepts regarding multiple hypothesis testing in a comprehensible language with well-illustrated examples. The frequently used adjustment tools, including the Bonferroni, the Holm-, the Hochberg-corrections, FDR, and q-value calculations, are implemented into our online calculator accessible at www.multipletesting.com (data are available upon request). The operation of the platform is intuitive, and sample data illustrate the underlying commands. Our goal for the future is to extend our current repertoire with new methodologies using auxiliary information (covariates). We believe our tool will be a valued resource for life science researchers and fills the gap, especially for those with limited programming experience.

## Supporting information

**S1 Script.**
(PHP)

**S2 Script.**
(PHP)

**S3 Script.**
(R)

## Author Contributions

**Conceptualization:** Balázs Győrffy.

**Data curation:** Balázs Győrffy.

**Formal analysis:** Balázs Győrffy.

**Funding acquisition:** Balázs Győrffy.

**Investigation:** Otília Menyhart.

**Methodology:** Otília Menyhart.

**Software:** Boglárka Weltz.

**Supervision:** Balázs Győrffy.

**Validation:** Boglárka Weltz, Balázs Győrffy.

**Visualization:** Otília Menyhart, Boglárka Weltz, Balázs Győrffy.

**Writing – original draft:** Otília Menyhart.

**Writing – review & editing:** Boglárka Weltz, Balázs Győrffy.

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
