## [Decision Letter · Decision Letter 0]

3 Mar 2021

PONE-D-21-00649

MultipleTesting.com: a tool for life science researchers for multiple hypothesis testing correction

PLOS ONE

Dear Dr. Menyhart,

Thank you for submitting your manuscript to PLOS ONE. After careful consideration, we feel that it has merit but does not fully meet PLOS ONE’s publication criteria as it currently stands. Therefore, we invite you to submit a revised version of the manuscript that addresses the points raised during the review process.

In the revised version of the paper please address the reviewers' comments listed at the end of this email. Additionally, please provide a comprehensive literature review in which you point out similar works in terms of tools development and works that use the type of analysis for which the current tool has been developed. Please carefully check the components presented in Figure 1. Also, please provide a real life science example in which the use of this tool will ease the work of the researches. Please provide the limitations of the study and point out the possible directions of extending the work. Finally, please compare the facilities offered by your tool with other similar tools available online.

We look forward to receiving your revised manuscript.

Kind regards,

Camelia Delcea

Academic Editor

PLOS ONE

Journal Requirements:

'The research was financed by the 2018-2.1.17-TET-KR-00001 and 2018-1.3.1-VKE-2018-00032 grants and by the Higher Education Institutional Excellence Programme (2020-4.1.1.-TKP2020) awarded to B. Gy, of the Ministry for Innovation and Technology in Hungary within the framework of the Bionic thematic program of the Semmelweis University. The funders had no role in study design, data collection and analysis, decision to publish, or preparation of the manuscript. The authors wish to acknowledge the support of ELIXIR Hungary (www.elixir-hungary.org).  '

We note that one or more of the authors are employed by a commercial company: A5 Genetics Ltd, Hungary

Reviewers' comments:

Reviewer's Responses to Questions

**Comments to the Author**

1. Is the manuscript technically sound, and do the data support the conclusions?

Reviewer #1: Yes

Reviewer #2: Yes

Reviewer #3: Yes

2. Has the statistical analysis been performed appropriately and rigorously? 

Reviewer #1: N/A

Reviewer #2: Yes

Reviewer #3: N/A

3. Have the authors made all data underlying the findings in their manuscript fully available?

Reviewer #1: Yes

Reviewer #2: No

Reviewer #3: Yes

4. Is the manuscript presented in an intelligible fashion and written in standard English?

Reviewer #1: Yes

Reviewer #2: Yes

Reviewer #3: Yes

5. Review Comments to the Author

Reviewer #1: The authors provide a review of the most frequently used approaches for correction methods in multiple hypothesis testing.

The simultaneous problem, introduced as early as the mid-twentieth century, is interesting and has been recently revived due to advanced in technology. However, I think the authors miss to give a contribution to the topic. There are several reviews on the subject which I believe are complete. [1], [2] and [3] are examples of extensive surveys over correction methods.

1. Stefanie R Austin, Isaac Dialsingh, and Naomi Altman. Multiple hypothesis testing: A review. J. Indian Soc. Of Agricultural Stat, 68:303–314, 2014.

2. Farcomeni A. A review of modern multiple hypothesis testing, with particular attention to the false discovery proportion. Statistical Methods in Medical Research, 17:347 (88), 2007.

3. Shaffer, J. P. (1995). Multiple hypothesis testing. Annual Review of Psychology, 46(1), 561–584.

Reviewer #2: The study “MultipleTesting.com: a tool for life science researchers for multiple hypothesis testing correction” is interesting. To facilitate multiple-testing corrections, the authors developed a fully automated solution not requiring programming skills or the use of a command line. The current research provides a much needed practical synthesis of basic statistical concepts regarding multiple hypothesis testing in a comprehensible language with well-illustrated examples. The web tool will fill the gap for life science researchers by providing a user-friendly substitute for command-line alternatives. The paper is well set, and the problem highlighted executed properly. However, attention should be given to the following highlighted points before resubmitting.

1. Page 10 of 22, “The procedure results in a statistical confidence measure, called a p-value, compared to the level of significance, α. When p is smaller than the confidence threshold α, the null hypothesis is rejected with a certain confidence, but the rejection does not "prove" the alternative hypothesis.” The P-value is called the observed significance level than why we are comparing with α. Why not take a straight level of significance from the P value. For example, if P = 0.11, means the significance level is 11%.

2. In the last, more recent references should be added to broaden the view of readers and enhance the new contribution of this paper for comparison.

3. The authors needed to add some more explanation in the conclusion section.

Reviewer #3: The authors developed an online tool for life science researchers for multiple hypothesis testing corrections. The online tool compiles the five most frequently used adjustment tools which can enable researchers to calculate False Discovery Rates (FDR) and q-values for multiple-testing corrections.

I suggest the author check the citation on page 9, paragraph 3, line 2 for proper citation.

I suggest moving the supplemental Table 1 to the main work due to its relevance.

Provide a friendly user-guide in the online tool to make it easier and more useful to targeted research communities.

The paper presented original research, reported in standard English with relevance literature discussed. With the level of contribution presented by the authors, I, therefore, recommend the manuscript for consideration by PLOS ONE after minor revision.

6. PLOS authors have the option to publish the peer review history of their article (what does this mean?). If published, this will include your full peer review and any attached files.

Reviewer #1: **Yes: **Antonino Abbruzzo

Reviewer #2: No

Reviewer #3: **Yes: **Badamasi Abba

---

## [Author Response · Author response to Decision Letter 0]

28 Apr 2021

Dear Editor-in-Chief,

Thank you for the opportunity to submit a revised version of our manuscript. 

Please find below our point-by-point responses to the issues raised by the reviewers (the implemented changes are also highlighted in the manuscript):

Comments from the Editor:

In the revised version of the paper please address the reviewers' comments listed at the end of this email. Additionally, please provide a comprehensive literature review in which you point out similar works in terms of tools development and works that use the type of analysis for which the current tool has been developed. Please carefully check the components presented in Figure 1. Also, please provide a real life science example in which the use of this tool will ease the work of the researches. Please provide the limitations of the study and point out the possible directions of extending the work. Finally, please compare the facilities offered by your tool with other similar tools available online.

Thank you for your precious input. We have addressed every comment raised by the reviewers and have extended our literature review by recent articles and tools developed for controlling multiplicity in high throughput data, a particularly rapidly growing field of science. We have substantially expanded the number of cited articles. We have also addressed the strategy illustrated in Figure 1. We are discussing examples for real-life applications, where controlling for multiplicity is a common issue. 

The topic of multiple hypothesis testing is far-reaching and rapidly proliferating, and our manuscript is limited to the introduction of the most extensively utilized methods. Another limitation is the lack of detailed description of technical parts of the presented methods, although the provided examples clarify the concepts. We articulate our goals more clearly in the revised version of the manuscript. 

We have conducted an extensive literature search but could not locate a similar user-friendly, web-based tool for conducting multiple hypothesis testing without the need for programming knowledge. We now include a section about tools implemented in R, and emphasize the facilities enabled by our platform. Our goal for the future is to extend the repertoire of available methods further, which we also highlight in the manuscript. 

Reviewer comments:

Reviewer #1: The authors provide a review of the most frequently used approaches for correction methods in multiple hypothesis testing.

The simultaneous problem, introduced as early as the mid-twentieth century, is interesting and has been recently revived due to advanced in technology. However, I think the authors miss to give a contribution to the topic. There are several reviews on the subject which I believe are complete. [1], [2] and [3] are examples of extensive surveys over correction methods.

1. Stefanie R Austin, Isaac Dialsingh, and Naomi Altman. Multiple hypothesis testing: A review. J. Indian Soc. Of Agricultural Stat, 68:303–314, 2014.

2. Farcomeni A. A review of modern multiple hypothesis testing, with particular attention to the false discovery proportion. Statistical Methods in Medical Research, 17:347 (88), 2007.

3. Shaffer, J. P. (1995). Multiple hypothesis testing. Annual Review of Psychology, 46(1), 561–584.

Thank you very much for the valuable suggestions. Following the reviewer's advice, we incorporated a section about additional, more recent methods, focusing mainly on novel FDR-controlling strategies. We have also included some of the suggested publications, complemented by additional comprehensive reviews, to enhance our contribution to the topic; please see pages 10-12.

Also, after reviewing the scientific literature for procedures and tools available for multiple testing, we could only identify tools requiring programming skills and the use of a command line. Although programming is increasingly important, this knowledge is not yet a default state among scientists. We feel that our online calculator accessible at www.multipletesting.com is our main contribution and fills a gap by not only providing a user-friendly option for researchers but also directly enabling the comparison of the results of the most widely used methods.

Reviewer #2: The study "MultipleTesting.com: a tool for life science researchers for multiple hypothesis testing correction" is interesting. To facilitate multiple-testing corrections, the authors developed a fully automated solution not requiring programming skills or the use of a command line. The current research provides a much needed practical synthesis of basic statistical concepts regarding multiple hypothesis testing in a comprehensible language with well-illustrated examples. The web tool will fill the gap for life science researchers by providing a user-friendly substitute for command-line alternatives. The paper is well set, and the problem highlighted executed properly. However, attention should be given to the following highlighted points before resubmitting.

Thank you very much for the positive remarks regarding our manuscript.

1. Page 10 of 22, "The procedure results in a statistical confidence measure, called a p-value, compared to the level of significance, α. When p is smaller than the confidence threshold α, the null hypothesis is rejected with a certain confidence, but the rejection does not "prove" the alternative hypothesis." The P-value is called the observed significance level than why we are comparing with α. Why not take a straight level of significance from the P value. For example, if P = 0.11, means the significance level is 11%.

Thank you for your suggestion. P-values indicate how extreme the data are, while alpha values determine how extreme the data must be before the null hypothesis is rejected. We have extended our description and clarified the concept further; please see page 4. We feel that our description helps the readers by showing the difference between these two concepts. 

2. In the last, more recent references should be added to broaden the view of readers and enhance the new contribution of this paper for comparison.

Thank you for the suggestion. According to the reviewer's request, we have incorporated additional work to enhance our contribution to the topic and to broaden the view of our readers, focusing mainly at recent FDR-controlling methods; please see pages 10-12.

3. The authors needed to add some more explanation in the conclusion section.

Thank you for your observation. We have incorporated additional examples, mainly aimed at life scientists, and expanded our Conclusions section; please see pages 12-13.

Reviewer #3: The authors developed an online tool for life science researchers for multiple hypothesis testing corrections. The online tool compiles the five most frequently used adjustment tools which can enable researchers to calculate False Discovery Rates (FDR) and q-values for multiple-testing corrections.

I suggest the author check the citation on page 9, paragraph 3, line 2 for proper citation.

Thank you for your remark, we have corrected the citation; please see page 9. 

I suggest moving the supplemental Table 1 to the main work due to its relevance.

Thank you for your suggestion; we now moved Table 1 to the main body of the manuscript.

Provide a friendly user-guide in the online tool to make it easier and more useful to targeted research communities.

Thank you for your recommendation. We have incorporated a short description into the online platform to enhance its practicality.

The paper presented original research, reported in standard English with relevance literature discussed. With the level of contribution presented by the authors, I, therefore, recommend the manuscript for consideration by PLOS ONE after minor revision.

We appreciate the positive evaluation and have edited the manuscript to increase its standards further.

We hope that the issues raised about the manuscript have been sufficiently addressed in this improved version. On this occasion, we would also like to thank the Editor and the three anonymous reviewers for their expert and helpful comments.

With best regards:

Balázs Győrffy MD PhD

---

## [Decision Letter · Decision Letter 1]

18 May 2021

MultipleTesting.com: a tool for life science researchers for multiple hypothesis testing correction

PONE-D-21-00649R1

Dear Dr. Menyhart,

We’re pleased to inform you that your manuscript has been judged scientifically suitable for publication and will be formally accepted for publication once it meets all outstanding technical requirements.

Kind regards,

Camelia Delcea

Academic Editor

PLOS ONE

Additional Editor Comments (optional):

Reviewers' comments:

Reviewer's Responses to Questions

**Comments to the Author**

1. If the authors have adequately addressed your comments raised in a previous round of review and you feel that this manuscript is now acceptable for publication, you may indicate that here to bypass the “Comments to the Author” section, enter your conflict of interest statement in the “Confidential to Editor” section, and submit your "Accept" recommendation.

Reviewer #2: All comments have been addressed

Reviewer #3: All comments have been addressed

2. Is the manuscript technically sound, and do the data support the conclusions?

Reviewer #2: Yes

Reviewer #3: Yes

3. Has the statistical analysis been performed appropriately and rigorously? 

Reviewer #2: Yes

Reviewer #3: Yes

4. Have the authors made all data underlying the findings in their manuscript fully available?

Reviewer #2: Yes

Reviewer #3: Yes

5. Is the manuscript presented in an intelligible fashion and written in standard English?

Reviewer #2: Yes

Reviewer #3: Yes

6. Review Comments to the Author

Reviewer #2: (No Response)

Reviewer #3: The authors have addressed all the reviewer's comments and notable improvement is observed in the revised version.

7. PLOS authors have the option to publish the peer review history of their article (what does this mean?). If published, this will include your full peer review and any attached files.

Reviewer #2: No

Reviewer #3: No

---

## [Editor Report · Acceptance letter]

1 Jun 2021

PONE-D-21-00649R1 

MultipleTesting.com: a tool for life science researchers for multiple hypothesis testing correction 

Dear Dr. Menyhart:

I'm pleased to inform you that your manuscript has been deemed suitable for publication in PLOS ONE. Congratulations! Your manuscript is now with our production department. 

Kind regards, 

on behalf of

Dr. Camelia Delcea 

Academic Editor

PLOS ONE